# Simultaneous Control and Guidance of an AUV Based on Soft Actor–Critic

**DOI:** 10.3390/s22166072

**Published:** 2022-08-14

**Authors:** Yoann Sola, Gilles Le Chenadec, Benoit Clement

**Affiliations:** 1Lab-STICC UMR CNRS 6285, ENSTA Bretagne, 29200 Brest, France; 2CROSSING IRL CNRS 2010, Flinders University, Adelaide 5005, Australia

**Keywords:** autonomous underwater vehicle, control, guidance, deep reinforcement learning, waypoint tracking, soft actor–critic, proportional integral derivative

## Abstract

The marine environment is a hostile setting for robotics. It is strongly unstructured, uncertain, and includes many external disturbances that cannot be easily predicted or modeled. In this work, we attempt to control an autonomous underwater vehicle (AUV) to perform a waypoint tracking task, using a machine learning-based controller. There has been great progress in machine learning (in many different domains) in recent years; in the subfield of deep reinforcement learning, several algorithms suitable for the continuous control of dynamical systems have been designed. We implemented the soft actor–critic (SAC) algorithm, an entropy-regularized deep reinforcement learning algorithm that allows fulfilling a learning task and encourages the exploration of the environment simultaneously. We compared a SAC-based controller with a proportional integral derivative (PID) controller on a waypoint tracking task using specific performance metrics. All tests were simulated via the UUV simulator. We applied these two controllers to the RexROV 2, a six degrees of freedom cube-shaped remotely operated underwater Vehicle (ROV) converted in an AUV. We propose several interesting contributions as a result of these tests, such as making the SAC control and guiding the AUV simultaneously, outperforming the PID controller in terms of energy saving, and reducing the amount of information needed by the SAC algorithm inputs. Moreover, our implementation of this controller allows facilitating the transfer towards real-world robots. The code corresponding to this work is available on GitHub.

## 1. Introduction

Controlling a robotic platform in a marine environment is a particularly challenging task since it is a hostile environment. It is strongly unstructured, meaning that the lack of structure makes the environment difficult to model. Moreover, this environment includes many uncertainties and external disturbances that cannot be easily predicted or modeled: the wind, the waves on the surface, the ocean currents, the seabed topography, the potential presence of objects, fishes, rocks, etc. Another specific problem found in marine robotics is the lack of positioning since the GPS signals cannot be propagated in the water. Without a valid or accurate model of the environment, the control task is more difficult and the controllers become harder to tune. Finally, as in many other control tasks, all of the input, output, and state signals include random noises, due to the environment or even the robot itself.

Robotics is a vast domain; mobile robotics is one of its subfields [1], where the studied robots are able to move by themselves, either on the ground, in the air, or in the water. Therefore, marine robotics [2] is itself a subfield of mobile robotics [3]. Various robotic platforms can be the objects of marine robotics [4], such as autonomous underwater vehicles (AUVs) [5], remotely operated underwater vehicles (ROVs) [6], unmanned surface vehicles (USVs) [7], autonomous sailboats [8], and underwater gliders [9].

In this paper, we only focus on AUVs as the other marine robots have different scientific issues. The development of AUVs began in the 1950s and has not stopped evolving; scientists are continuously improving the mechanical designs, actuators, sensors, electronics, communication, power supply, and control algorithms. Their uses have also diversified, i.e., from research (hydrography, oceanography) and military applications (communication/navigation network nodes, mine countermeasures), to commercial and hobby uses.

In this work, we compare a machine learning-based controller (the SAC algorithm) and a classical controller (the PID controller) on a waypoint tracking task performed by an AUV. This control task can be generalized to a large number of marine robotics missions since every path can be decomposed into successive waypoints to follow. Every mission where trajectories need to be followed or specific targets need to be reached can be reduced to a waypoint tracking task composed of one or more waypoints.

## 2. Related Works

The use of machine learning and its subfields (deep learning, reinforcement learning, and deep reinforcement learning, which will be detailed in the following sections) applied to robotics and control theory has grown exponentially in the past few years. Machine learning can either replace or improve the control algorithms used in robotics: **its modeling power and its ability to generalize to new events or behaviors can overcome the limits of certain control approaches**.

Several surveys on deep learning applied to robotics can be found in the literature, such as [10,11,12,13] (in chronological order).

Reference [14] provided an interesting discussion about the limits and the potentials of deep learning approaches for robotics. The authors mainly dealt with the learning challenges, robotic vision challenges, and reasoning challenges (the inferences or conclusions generated by the processing of any input information). **The authors detailed the need for better evaluation metrics**, as well as better simulations for robotic vision. They reviewed the perception, planning, and control in robotics, and concluded by saying that neural networks generally do not perform well when the state of the robot falls outside the training dataset. The work also defined the concepts of programming and data as spectra, allowing to automatically derive deep learning algorithms from reasonable amounts of data and suitable priors.

Reference [15] presented a large comparative study of reinforcement learning applied to control theory. A controller based on neural networks and reinforcement learning was tested on several systems, i.e., an AUV, a plane, the magnetic levitation of a steel ball, and the heating coil (belonging to the set of heating, ventilation, and air conditioning (HVAC) problems). These benchmarks allowed the evaluation of several control theory aspects: **the effects of nonlinear dynamics**, the reaction to varying setpoints, long-term dynamic effects, **the influence of external variables** and the evaluation of the precision.

Reference [16] presented a survey on reinforcement learning applied to robotics (it is a good entry point). However, this article was published in 2013 and the deep reinforcement learning (DRL) approaches were never mentioned since the majority of modern DRL algorithms were not yet published at that time.

In the literature, we can find many works dealing with machine learning applied to the control of unmanned aerial vehicles (UAV) since UAVs have one of the largest communities of researchers in the mobile robotics field.

Reference [17] presented an interesting review on DRL applied to UAVs. The authors divided UAV tasks into several categories: path planning (using real-time images, with or without ground maps), navigation (for discrete or continuous action spaces), and control (for attitude control, longitudinal and lateral control, image processing, or UAV swarm control).

Reference [18] presented a review on a more specific subject: the use of UAVs for obstacle detection and collision avoidance in the Internet of Things (IoT) field. They provided many useful resources, such as a list of available datasets or different hardware and communication architectures. The review mainly listed convolutional neural network (CNN) works applied in a number of industrial applications. The UAVs controlled in these industrial environments need to be aware of the possible collisions with workers, mobile vehicles, robots, or even heavy-duty tools.

Reference [19] used reinforcement learning to control the attitudes of UAVs. Both works implemented PPO, TRPO, and DDPG algorithms for this task, and compared them to a PID controller. **In our work, we also compare RL algorithms to PID controllers since it is one of the most used controllers in the literature**.

Reference [20] is a good example of reinforcement learning applied to the navigation of a UAV. **An exact mathematical model of the environment is not always available in unknown environments**, so this work applied Q-learning to indicate the position of a UAV based on the previous position, by learning its own model of the environment. This information was then given to a PID controller.

Some research studies deal with higher levels of control. For example, Reference [21] used the SAC algorithm to directly generate trajectories for UAVs. The authors focused their efforts on generating trajectories—allowing to save energy—since their use case was a data collection system based on UAVs. They wanted to optimize the UAV trajectory to minimize the time required to complete the task, which ensured that the robot used less energy. **SAC is the algorithm we implemented in our work, which further confirms that it can be successfully applied to mobile robots, such as UAVs or AUVs**.

Reference [22] presented another example of a deep reinforcement learning algorithm used to make UAVs energy-efficient. The use case studied in this paper involved the control of a group of UAVs to create a communication network. The DDPG algorithm was used to make the whole group energy-efficient and to make the communication better (in terms of coverage and fairness). **This work shows that machine learning can help to save energy when ML algorithms are used as the controllers of the robots. Energy management is one of the main challenges of robotics**.

Finally, Reference [23] presented a recent survey that studied the machine learning methods applied to groups or flocks of UAVs. The authors listed approaches taken from the three main subfields of machine learning: supervised learning, unsupervised learning, and reinforcement learning.

More researchers are experimenting with implementing machine learning algorithms inside autonomous underwater vehicle (AUV) systems. As said in the introduction, the marine environment has its own constraints, which differ from the aerial and terrestrial environments; thus, the use of machine learning techniques can help exceed these difficulties.

Reference [24] is an example of deep learning applied to AUVs: A neural network was used to perform an AUV trajectory tracking task using a neural network control approach. Two neural networks (NNs), called the *actor* and the *critic*, were implemented based on the AUV model derived in the discrete-time domain. The critic was used to evaluate the long-time performance of the designed control, while the actor compensated for the unknown external disturbances.

Reference [25] used a neural network to tune the PID controller of an AUV in real-time. The neural network was implemented in parallel to the PID to perform an auto-tuning control of the AUV. This interesting approach was able to mix both control theory and machine learning techniques inside the same AUV controller.

Since many reinforcement learning (RL) algorithms are more appropriate to continuous control tasks [26], most of the machine-learning-based controllers for AUV fall into this category.

Many works can be found for the low-level control of AUVs. Reference [27] involved an early trial, allowing to control the AUV thrusters in response to command and sensor inputs. The authors used a Q-learning approach based on a neural network, which was a rare instance of DRL at that time (in 1999).

The RL controller created in [28] was robust to thruster failures. It was based on model-based evolutionary methods. The problem was modeled by a Markov decision process (MDP) and the controller was based on a parametrized policy updated by a direct policy search method. The controller was able to operate under-actuated AUVs with fully or partially broken thrusters.

References [29,30] both implemented the DDPG algorithm to control an AUV. The first paper used it to create a depth controller, allowing track desired depth trajectories, while the second paper allowed the AUV to follow linear velocities and angular velocity reference signals.

Reference [31] proposed a specific reward function design to perform AUV docking tasks. The authors tested their reward function formulation by comparing its implementation inside the PPO, TD3, and SAC algorithms, and managed to achieve successful results.

**In many of these low-level control problems, the input vectors given to the RL controllers were often composed of many variables to correctly follow the reference signals given by the guidance components. The reward must be specifically designed for each control task**.

In the literature, we also found papers applying RL to the guidance or high-level control of AUVs. **These approaches are often able to replace both the control and guidance components of the systems, but this is not systematic**.

Reference [32] implemented a classic actor–critic architecture to carry out the waypoint tracking and obstacle avoidance tasks of an AUV. Reference [33] was also able to make an AUV fulfill the ‘path following’ and ‘collision avoidance’ missions but using a PPO algorithm.

Reference [34] performed a trajectory tracking task of an AUV using the DPG algorithm and recurrent neural networks. The motion control was only conducted in a 2D horizontal plane. It compared this method with a PID controller and other non-recurrent methods.

Reference [35] was a rather creative work since it used the DDPG algorithm to plan the trajectories of multiple AUVs. The goal was to estimate a water parameter field inside an under-ice environment.

## 3. Materials

In this section, we expose the theoretical elements of the implemented algorithms.

### 3.1. Elements of the Control Theory

We first present the control theory elements; more precisely, the proportional integral derivative (PID) controller is our baseline on the waypoint tracking task.

#### 3.1.1. Guidance–Navigation–Control Systems

In control theory, a *guidance–navigation–control (GNC)* system is the name of the global hierarchy of algorithms structuring the control of a dynamical system [36]. A general GNC system is represented in Figure 1. Each block represents an independent system or algorithm. The components of a GNC system are the following:
**The plant system:** The plant is the system that needs to be controlled. Its output signals *y* are the variables being controlled. The plant system used in this work will be an autonomous underwater vehicle (AUV).**The navigation algorithm:** The role of the navigation component is to estimate the output *y* of the plant system named $\hat{y}$. Sometimes the output cannot be directly determined and an algorithm is therefore required to perform the estimation, i.e., when the measures are noisy (filtering) or when some information is missing (interpolation), etc. This component is also called a *state observer* and is said to perform a *state estimation* [37]. The extended Kalman filter [38], regression techniques [39], the Bayesian estimation [40], or the interval analysis [41] are common examples of navigation approaches.**The control algorithm:** Often called the controller; its function computes the input *u* of the plant [42]. This input is based on the difference between the estimate $\hat{y}$ of the output (given by the navigation component) and a reference *r* (given by the guidance component). This component can also be referred to as the *low-level controller*. The main control theory approaches are the classical control theory [43], adaptive control [44], stochastic control [45], robust control [46], optimal control [47], fuzzy logic [48], hierarchical control system (HCS) [49], and nonlinear control [50].**The guidance algorithm:** The guidance component is in charge of generating the reference *r*. It takes into account the current estimate of the output, $\hat{y}$, and several parameters *p* specified in advance (the parameters can be either variable or fixed). The guidance algorithm allows for fulfilling different goals [1], e.g., path planning [51], obstacle avoidance [52], waypoint tracking [53], etc. These different goals can be combined during the same control task, and a trade-off must be found between each of them. Line-of-sight techniques [54], artificial potential field methods [55], and Voronoi diagrams [56] are examples of well-known guidance approaches. The guidance system can also be called the *high-level controller*.

Each component of a GNC system can be considered a separate dynamical system, with its own inputs, outputs, and state variables. All of the components can also be reduced to one dynamic system encapsulating all of them, depending on the granularity needed for the use case.

We focus our study on the guidance and control algorithms, assuming that the navigation problem is resolved.

#### 3.1.2. PID Controllers

The proportional integral derivative (PID) controller is the most used controller in both the academic field and the industry [57,58] and belong to the classical control theory approaches.

The input *u* needed by the plant system is computed by the PID thanks to the tracking error $e\left(t\right)=r\left(t\right)-\hat{y}\left(t\right)$, with r(t) being the reference and $\hat{y}\left(t\right)$ being the estimated output of the plant. The equation used by the PID controller is the following:(1)$$u\left(t\right)\;=\;k_{p}\;e\left(t\right)\;+\;k_{i}\;\int_{0}^{t}e\left(\tau \right)d\tau \;+\;k_{d}\;\frac{de(t)}{dt}$$
where $k_{p}$, $k_{i}$, and $k_{d}$ are real-value scalars or matrices in the case of multi-dimensional signals of multi-input multi-output (MIMO) systems, called the *gains*.

As shown in Equation (Equation 1), the PID controller takes its name from the fact that it computes the input *u* proportionally to three distinct terms—the tracking error (representing the past), the integral of the tracking error (representing the present), and the derivative of the tracking error (representing the future). The block diagram in Figure 2 shows a typical PID controller in the frequency domain (with the use of Laplace transforms), without guidance and navigation components.

The gains $k_{p}$, $k_{i}$, and $k_{d}$ affect, respectively, the speed of convergence (the bigger $k_{p}$ is, the faster the system converges to the reference), the static error (the difference between the input and the output of a system when the time converges to the infinity), and the magnitude of the oscillations (the bigger $k_{d}$ is, the smaller the oscillations of the output are).

These gains can either be constant (fixed by the control theory methods [59,60]) or variable. In the latter case, they will be updated automatically during the execution of the control task according to a criterion or an external algorithm [61,62]. They can also be tuned using the frequency domain [63].

The PID controller is easy to implement, given that only Equation (Equation 1) is needed. It is also easy to tune since it has just three parameters, which can eventually be tuned empirically by interpreting the resulting signals. However in some cases, even a self-tuning PID controller is not enough, and advanced approaches may be considered. Several limitations can be found in the PID control approach [64,65]. A PID controller can have difficulties with the systems involving specific non-linearities or varying internal parameters. It can also lack responsiveness in the presence of large low-frequency disturbances. Finally, the tuning of the PID controller must be carefully chosen since a trade-off must be found between the regulation abilities and the response time of the controller.

### 3.2. Elements of Deep Reinforcement Learning

We are now going to introduce several principles of deep reinforcement learning (DRL). DRL is a subfield of machine learning mixing approaches taken from reinforcement learning (RL) and deep learning (DL). In this section, we mainly present the paradigm used in RL as well as the specificities of the soft actor–critic algorithm.

#### 3.2.1. Reinforcement Learning

In reinforcement learning (RL), problems are defined using the paradigm of the Markov decision process (MDP) [66]. A MDP is composed of an agent attempting to achieve a goal defined by the task. The agent evolves in an environment and learns the task by trial and error.

A MDP is represented on the Figure 3. At time step *t*, the agent is in a given state $S_{t}$ of the environment and performs an action $A_{t}$. The state and the action can be the vectors. The environment responds to the agent by sending to it an observation of the new state $S_{t+1}$ and a scalar signal called the *reward*
$R_{t+1}$. The role of the reward is to judge the actions carried out by the agent with respect to the states of the environment. If the reward is positive, the action is good to carry out in the given state to complete the task; otherwise, it is a bad action to perform and it moves the agent away from its goal. The tuple $\left(S_{t},A_{t},S_{t+1},R_{t+1}\right)$ is called a *transition* in the vocabulary of MDPs.

The behavior of the agent is defined by the policy π(a|s), a function allowing the agent to choose which action to perform in a given state. The policy can either be stochastic or deterministic. Deterministic policies generate a single action π(s)=a, while stochastic policies generate a vector composed of the probabilities of choosing each possible action, also called the probability distribution over actions $\pi \left(a|s\right)=P_{\pi }\left[A=a|S=s\right]$.

In RL tasks, the goal of the agent is to maximize the total sum of discounted rewards, called the *return* $G_{t}$ (sometimes also called the *discounted future reward*):(2)$$G_{t}=R_{t+1}+\gamma R_{t+2}+\cdots =\sum_{k=0}^{\infty }\gamma^{k}R_{t+k+1}$$
where $R_{t}$ is the reward received at the time step *t* and γ is the discount factor belonging to [0,1], allowing to control the influence of the future expected rewards, which may be estimated with large uncertainties.

The value functions are specific functions used by the agent to assess which states and/or actions are the best to maximize the return based on the current policy π. To achieve a RL task, the agent has to correctly learn the policy and the value functions based on the information sent by the environment.

The *state-value function* $V_{\pi }\left(s\right)$ is defined by the expected return computed from a given state *s*, following a given policy π:(3)$$V_{\pi }\left(s\right)=E_{\pi }\left[G_{t}|S_{t}=s\right]$$

The *action-value function* $Q_{\pi }\left(s,a\right)$, also called the *Q-value function* or the *Q-function*, is defined by the expected return computed from a given action *a* taken in a given state *s*, following a given policy π:(4)$$Q_{\pi }\left(s,a\right)=E_{\pi }\left[G_{t}|S_{t}=s,A_{t}=a\right]$$

The RL algorithms can be either model-free (the agent does not have access to a model of the environment) or model-based (the agent has access to a model of the environment, and in some tasks, it also has to learn this model). In both cases, the agent needs to face a well-known issue in RL: the exploration–exploitation trade-off.

Indeed, the agent needs to explore the environment by interacting with it to discover the action and the state spaces, which will provide the information needed for estimating the value and policy functions (and for the model of the environment in some cases). In the same way, the agent needs to exploit the already gathered knowledge to choose the right actions to maximize the return, which will allow the agent to fulfill the RL task.

On the one hand, if the agent spends too much time exploring the environment, it will never fulfill the initial task. On the other hand, if it does not explore enough, the estimates of its value functions and its policy will not be accurate enough to choose the right actions to maximize the return, and again, the agent will not fulfill the task. An appropriate exploration strategy needs to be implemented by the agent to resolve this trade-off.

If the task is supposed to never end (e.g., an inverted pendulum equipped with a motor, attempting to balance itself indefinitely), it is called a *continuing task*. If the task is supposed to end when the agent reaches a terminal state $S_{T}$ (whenever it succeeds or fails the task), it is called an *episodic task*. The task is then broken down into *episodes*, and each episode is composed of the same amount of maximal time steps. Each time the agent ends an episode with a success or a failure, a new one begins, with (or without) a new configuration.

When the task is being learned, the parameters of the policy and the value functions are iteratively updated; we are referring to *training episodes*. When the learning process is over, the parameters of the policy and the value functions are fixed; we are referring to *testing episodes*. The testing episodes allow evaluating the performance of the policy in real conditions, so no rewards are given to the agent during this phase.

#### 3.2.2. Soft Actor–Critic

The soft actor–critic (SAC) algorithm is one of the most recent DRL techniques [67]. It is an evolution of the deep deterministic policy gradient (DDPG) algorithm [26] and takes ideas from the trust region policy optimization (TRPO) [68] and proximal policy optimization (PPO) [69] approaches. Additional improvements have been made to SAC [70] to implement elements from the twin-delayed deep deterministic (TD3) algorithm [71]. The features of the SAC algorithm are the following:**Deep reinforcement learning:** It is a subfield of ML where the learning tasks are formulated using the Markov decision process paradigm (reinforcement learning), and where the policy and the value functions are modeled using neural networks (deep learning).**Model-free:** SAC does not need a model of the environment to learn the task.**Stochastic policy:** The policy implemented by a neural network is not deterministic. The outputs of the policy network correspond to probabilities over the possible actions.**Policy gradient:** The policy is parameterized and is learned iteratively based on gradient computations. It is opposed to *value-based* methods, where the policy is derived by simply taking the maximum Q-values. The policy gradient theorem [72] has been developed to simplify the computations of the gradients needed for the update of the policy’s parameters.**Actor–critic:** It is a RL architecture where an *actor* has to learn the policy of the agent, while a *critic* learns the value functions. Both components learn their respective functions simultaneously, and communicate with each other to improve their estimations. The output of the critic allows improving the actor, and the critic is updated based on the actions chosen by the actor.**Off-policy:** The policy can be updated based on observations gathered from an older iteration of itself, or even by another policy. The exploration policy can be different from the true policy learned by the agent (called the target policy).In the SAC algorithm, an *experience replay* mechanism is implemented: The transitions $\left(S_{t},A_{t},S_{t+1},R_{t+1}\right)$ observed by the agent are successively stored inside a replay buffer; this replay buffer is then randomly sampled to update the neural networks of the actor and the critic. This is a mechanism often used in off-policy algorithms.

The adjective *soft* in the name *soft actor–critic* comes from the fact that an additional term called the entropy H(.) is added to the objective function maximized by the agent, in order to encourage exploration. The entropy is a measure of the predictability of an agent. The more the policy of the agent is certain of which action is the best to obtain the highest cumulative reward in a given state, the lower the entropy of the policy will be. In other words, the lower the entropy is, the more deterministic the policy will be. It is defined as:(5)$$H\left(\pi_{\theta }\left(.|s_{t}\right)\right)=E_{a∼\pi_{\theta }\left(.|s\right)}\left[-\log\left(\pi_{\theta }\left(a|s\right)\right)\right]$$

The agent is trained with the objective of maximizing both the expected return $G_{t}$ (Equation 2) and the entropy. Throughout the training episodes, the agent needs to maximize the following objective function:   
(6)$$J\left(\theta \right)=\sum_{t=0}^{T}\gamma^{t}\;E_{\left(s_{t},a_{t}\right)∼\rho_{\pi }}\left[r\left(s_{t},a_{t}\right)+\alpha H\left(\pi_{\theta }\left(.|s_{t}\right)\right)\right]$$
where α is a hyperparameter controlling how important the entropy will be (called the *temperature*), γ is the discount factor defined in previous sections, and $\rho_{\pi }$ is the state and state-action marginals of the trajectory distribution induced by the policy $\pi_{\theta }\left(a|s\right)$ [67]. The temperature α must not be confused with the learning rate, allowing to update the neural networks (they will be noted differently here to avoid confusion).

The goal of maximizing the entropy is to learn a policy that acts as randomly as possible to encourage exploration, while still being able to succeed at the task (exploitation). This allows avoiding situations in which the agent might fall into a local optimum behavior. Moreover, maximizing the entropy can help to capture multiple modes of near-optimal strategies. If there exist multiple options that seem to be equally good, the policy should assign each with an equal probability to be chosen. This maximum-entropy policy can also give more robustness to the agent, allowing it to be more robust to abnormal or rare events occurring during the task. The SAC algorithm is the follow-up to the soft Q-learning algorithm [73], created by the same authors.

The SAC algorithm uses neural networks to learn three functions:The policy $\pi_{\theta }$ modeled by a neural network with the parameters θ.The soft Q-value function $Q_{w}$ modeled by a neural network with the parameters w corresponding to the Q-value function derived from the new entropy-regularized reward.The soft state-value function $V_{\psi }$ (or sometimes simply called soft value function) modeled by a neural network with the parameters ψ corresponding to the state-value function derived from the new entropy-regularized reward.A fourth neural network $V_{\psi^{\prime }}$, modeled with the parameters $\psi^{\prime }$, and called the target network, is used to stabilize the learning process. It is updated slowly using a moving average on the parameters of the soft state-value function:
(7)$$\psi^{\prime }\leftarrow \tau \psi +\left(1-\tau \right)\psi^{\prime }\;\;\;\;\;\;\;\mathrm{with}\;\;\tau \ll 1$$

The neural network of the soft state-value function is trained to minimize the objective function [67]:(8)$$\begin{array}{cc} J_{V}\left(\psi \right) & =E_{s_{t}∼D}\left[\frac{1}{2}{\left(V_{\psi }\left(s_{t}\right)-E\left[Q_{w}\left(s_{t},a_{t}\right)-\log\pi_{\theta }\left(a_{t}|s_{t}\right)\right]\right)}^{2}\right]\\ \mathrm{with}\;\mathrm{gradient}:\;\nabla_{\psi }J_{V}\left(\psi \right) & =\nabla_{\psi }V_{\psi }\left(s_{t}\right)\left(V_{\psi }\left(s_{t}\right)-Q_{w}\left(s_{t},a_{t}\right)+\log\pi_{\theta }\left(a_{t}|s_{t}\right)\right) \end{array}$$
where D is the replay buffer.

The neural network of the soft Q-value function is trained to minimize the objective function [67]:(9)$$\begin{array}{cc} J_{Q}\left(w\right) & =E_{(s_{t},a_{t})∼D}\left[\frac{1}{2}{\left(Q_{w}\left(s_{t},a_{t}\right)-\left(r\left(s_{t},a_{t}\right)+\gamma E_{s_{t+1}∼\rho_{\pi }\left(s\right)}\left[V_{\psi^{\prime }}\left(s_{t+1}\right)\right]\right)\right)}^{2}\right]\\ \mathrm{with}\;\mathrm{gradient}:\;\nabla_{w}J_{Q}\left(w\right) & =\nabla_{w}Q_{w}\left(s_{t},a_{t}\right)\left(Q_{w}\left(s_{t},a_{t}\right)-r\left(s_{t},a_{t}\right)-\gamma V_{\psi^{\prime }}\left(s_{t+1}\right)\right) \end{array}$$

Similar to TRPO, the SAC algorithm uses the Kullback–Leibler (KL) divergence [74,75] to quantify the similarity between the policy before and the policy after the update of its parameters. The actor minimizes the following objective function to update the policy network:(10)$$\begin{array}{cc} J_{\pi }\left(\theta \right) & =D_{\mathrm{KL}}\left(\pi_{\theta }\left(.|s_{t}\right)∥\exp\left(Q_{w}\left(s_{t},.\right)-\log Z_{w}\left(s_{t}\right)\right)\right)\\ \; & =E_{a_{t}∼\pi }\left[\log\pi_{\theta }\left(a_{t}|s_{t}\right)-Q_{w}\left(s_{t},a_{t}\right)+\log Z_{w}\left(s_{t}\right)\right] \end{array}$$

The partition function $Z_{w}\left(.\right)$ is usually intractable, but this is not a problem in practice since it will not contribute to the gradient $\nabla_{\theta }J_{\pi }\left(\theta \right)$. Reference [67] presents a proof showing that this policy update will guarantee that $Q^{\pi_{\mathrm{new}}}\left(s_{t},a_{t}\right)\geq Q^{\pi_{\mathrm{old}}}\left(s_{t},a_{t}\right)$.

Finally, all of the networks can be updated multiple times in a row in only one step of the environmental sampling. The gradient descents use different learning rates $\alpha_{\pi }$, $\alpha_{Q}$, and $\alpha_{V}$ for updating the functions $\pi_{\theta }$, $Q_{w}$, and $V_{\psi }$, respectively. These learning rates and the weighting factor τ used for updating the target soft state-value network are hyperparameters that need to be tuned.

A simplified pseudocode of the SAC is given in the Algorithm 1:
**Algorithm 1:** Soft Actor-Critic (SAC)1.Random initialization of the parameter vectors θ, w, ψ.Initialization of the target parameters: $\psi^{\prime }\leftarrow \psi$2.**for** each iteration **do**
(a)**for** each environment step **do**
i.$a_{t}∼\pi_{\theta }\left(a_{t}|s_{t}\right)$ii.$s_{t+1}∼\rho_{\pi }\left(s_{t+1}|s_{t},a_{t}\right)$iii.$D\leftarrow D\cup \{\;\left(s_{t},a_{t},r\left(s_{t},a_{t}\right),s_{t+1}\right)\;\}$
(b)**for** each gradient update step **do**
i.$\psi \leftarrow \psi -\alpha_{V}\nabla_{\psi }J_{V}\left(\psi \right)$ii.$w\leftarrow w-\alpha_{Q}\nabla_{w}J_{Q}\left(w\right)$iii.$\theta \leftarrow \theta -\alpha_{\pi }\nabla_{\theta }J_{\pi }\left(\theta \right)$iv.$\psi^{\prime }\leftarrow \tau \psi +\left(1-\tau \right)\psi^{\prime }$



## 4. Methods

We will now detail our use case and methods to deal with it.

### 4.1. Our Simulation and Task Setups

In this section, we describe the tools we used to build our simulations, as well as the AUV model we chose and the control task we wanted to fulfill.

#### 4.1.1. The Architecture of Our Simulations

The simulated environment needs to be representative of the ground truth and has to offer rewards and appropriate state observations to the deep reinforcement learning agent.

Figure 4 explains how our simulation was structured. The main tools used for implementing our use case are:**ROS:** The robotic operating system (ROS) [76] is one of the most used types of middleware in robotics. It allows structuring an application in distinct components and managing all of the communications made between them. The different parts of the GNC system (see Section 3.1.1) are run independently by ROS and this middleware allows to easily transfer the control algorithms from a simulation to a real robotic platform.**Gazebo:** This is a popular robotics simulator [77] and is based on ROS. Gazebo can be launched from ROS or independently, and many interactions are available between ROS components and Gazebo. A graphics engine and a physics engine are provided by Gazebo. The physics engine mainly includes collision management, but many plugins can add new elements, such as aerodynamics or hydrodynamics.**UUV Sim:** The unmanned underwater vehicle simulator (UUV Sim) [78] is a marine robotics simulator focused on underwater robotic platforms. It is based on ROS and Gazebo and adds plugins, allowing the support of hydrodynamic forces, such as waves or ocean currents, as well as some marine robot models (mainly AUVs and ROVs) and environments. It also provides templates for control algorithms. The hydrodynamics forces and the AUV and ROV models are based on the Fossen model [79], which is the standard modeling approach in marine robotics.**PyTorch:** A Python library allowing to define neural network architectures [80]. It allows training these models using the graphics processing unit (GPU) found on computers, which helps to reduce the computation time. It was used to define the neural networks of the SAC algorithm, as well as for other support functions such as the real-time monitoring of the training process.**MATLAB:** We chose to analyze post-simulation results using MATLAB [81], a well-known computing tool.

Besides MATLAB, all tools were used inside ROS components. This is why the middleware ROS encapsulated all of the other tools in Figure 4, except MATLAB. ROS can interact with the UUV simulator during the execution of a simulation and can use a variety of features in real-time, e.g., receiving the state variables of the AUVs, changing the direction and the magnitude of the ocean currents, sending commands to the thrusters, simulating sensor failures, etc. This whole simulation setup can be reusable for a variety of tasks involving marine robotics and/or machine learning.

#### 4.1.2. The Autonomous Underwater Vehicle

For this work, we used the RexROV 2 model provided by the UUV simulator. The RexROV 2 is usually operated remotely by a human, but here it was considered an AUV since it acted autonomously due to control algorithms. The dimensions and parameters of the RexROV 2 were derived from model parameters of the SF 30k ROV [82]. As shown in Figure 5a, the RexROV 2 is a cube-shaped ROV. It has six degrees of freedom (6DOF), three possible translations, three possible rotations, and it is actuated due to six thrusters or propellers. These thrusters were placed according to an unconventional layout (see Figure 5b). One given thruster is able to impact multiple DOF at the same time, and each basic movement needs a specific combination of thrusters inputs. The RexROV 2 sensors are composed of an inertial measurement unit (IMU) measuring the linear and angular accelerations, velocities, and positions in the AUV frame, a Doppler velocity log (DVL) measuring the linear velocities of the AUV relative to the seabed, and a pressure sensor measuring the depth of the AUV. Those sensors have several measurements in common but they are coupled with each other to provide more accurate estimates (this is called *sensor fusion*). Having multiple sensors measuring the same variables also allows for redundancy in case of sensor failure. The robot can also have an RGB camera and a GPS sensor (used when the AUV is surfacing) but they were not implemented in this work. This particular AUV was chosen for its shape allowing for ease of controllability.

The notations used to describe the RexROV 2 variables are the following:(11)$$\left\{\begin{array}{ccc} \eta & = & {\left[x,y,z,\phi ,\theta ,\psi \right]}^{T}\\ \nu & = & {\left[v_{x},v_{y},v_{z},\omega_{x},\omega_{y},\omega_{z}\right]}^{T}\\ u & = & {\left[u_{1},u_{2},u_{3},u_{4},u_{5},u_{6}\right]}^{T} \end{array}\right.$$
where η is the vector composed of the position x=(x,y,z) (expressed in the Gazebo reference frame, and corresponding, respectively, to the surge, the sway, and the heave) and the Euler angles (roll, pitch, and yaw) Θ=(ϕ,θ,ψ), ν is the vector composed of the linear velocities $v=(v_{x},v_{y},v_{z})$ and the angular velocities $\omega =(\omega_{x},\omega_{y},\omega_{z})$, and u is the vector composed of the control and propulsion forces composed of the control inputs $u_{i}$ sent to the six thrusters of the RexROV 2.

#### 4.1.3. The Control Task

AUVs can be employed for various types of missions or control tasks, e.g., path planning, obstacle avoidance, waypoint tracking, station keeping, etc. In this work, we chose to make the RexROV 2 perform a waypoint tracking task, represented on Figure 6. The task was only operated in simulation for this work. This waypoint tracking problem aims to compare a deep-reinforcement-learning-based controller with a PID controller. Following the RL terminology, the task is divided into a finite number of episodes. In each episode, the AUV must reach a different 3D point, called the waypoint or target waypoint. These episodes are composed of time steps, and the maximum number of time steps has to be tuned for each RL task.

At the beginning of each episode, we initialized the AUV at the 3D position x=[0,0,−20] (in meters and relative to the frame attached to the Gazebo world center) and we assigned it a random orientation Θ. More precisely, the Euler angles were initialized with ϕ=θ=0 and ψ taken randomly in the range [0, 360].

At the beginning of each episode, a waypoint was randomly placed inside a bounded 3D box in the world of the simulator centered on the position [0,0,−20]. The 3D box takes the range [−20, 20] on both *x* and *y* axes and the range [−60, −1] on the *z* axis.

During the execution of a training or a testing episode, the AUV cannot exceed the boundaries of the 3D box. If it does, the episode ends with a failure called a *collision*. If the AUV is able to reach a distance of less than 3 m from the waypoint without going outside of the vertical boundaries, the episode ends with a success. Finally, if the AUV spends too much time steps without reaching the waypoint, the episode ends with a failure (we call this case of failure a *timeout*). We empirically set the maximum number of time steps to 1000, which is a sufficient duration for the AUV to reach the target waypoints.

To make the simulation more realistic, we added noise to the sensor measurements of all of the state variables of the RexROV 2. For each variable, a noise $\sigma_{i}$ was added to its original value by randomly sampling the interval [0.05, 0.1] following a uniform distribution. A random noise $\sigma_{i}$ was also added to each component of the action vector u by uniformly sampling the interval [0.01, 0.05].

We also added external disturbances to our underwater simulated environment thanks to the use of fluctuating ocean currents. Every 100 timesteps, the ocean current velocity $c_{v}\in \left[0,\;1\right]$ (in m·s${}^{-1}$) and the ocean current angles $\left(c_{ha};c_{va}\right)\in \left[-0.5,\;0.5\right]$ (for horizontal and vertical angles, respectively, in radians) were randomly modified by uniformly sampling new values in their respective ranges. Figure 7 shows an example of the ocean current vector in the Gazebo reference frame. The fact that the angles and the velocity of the ocean currents were constantly changing during the execution of the episodes added a real challenge to the control task.

The UUV simulator offers several underwater environment files, called *worlds*, which can be either fictional (empty underwater world, AUV underwater world, ocean waves world, lake) or based on real-world locations (Hercules shipwreck, the coast of Mangalia in Romania, Munkholmen in Norway, subsea BOP panel). We used the *Ocean waves world,* which is a generic underwater environment providing a realistic seabed with a great variability of reliefs.

This work focused on the low-level control of the AUV as well as the integration of guidance functions. The deep RL-based controller had to perform what is called an end-to-end control. This means that the navigation component of the GNC system (see Section 3.1.1) was kept relatively simple for this control task and was not subject to specific studies. The navigation component was provided by Gazebo, which simply sent the true values of the AUV variables with added noises.

Figure 5b shows the thruster layout of RexROV 2. We can see that some of its thrusters can affect multiple degrees of freedom (DOF) at the same time, because of their orientations. Thus, the DOF of this AUV are strongly correlated, making the control task even harder to perform. Other stable cube-shaped AUVs would be able to move horizontally or vertically using independent thrusters, but RexROV 2 required specific thruster combinations for each movement. Understanding the dynamics of this AUV represents a real challenge for any learning algorithm.

### 4.2. Our Implementation of the Control Algorithms

In this section, we describe the algorithms we used for the control of RexROV 2 as well as the metrics allowing us to judge the respective performances. The task described in Section 4.1.3 was performed by both the soft actor–critic (SAC) algorithm (see Section 3.2.2) and a PID controller (see Section 3.1.2) using the UUV simulator. During our work, we tested many different configurations for the SAC algorithm. We evaluated each configuration by making it perform a trial on the task, i.e., attempting to make it learn the task. For each trial, the procedure involved the following:1.**Training phase:** The SAC algorithm was trained on successive training episodes. Each episode replicated the task described in Section 4.1.3, with a randomly placed target waypoint and random time-varying ocean currents. The SAC-based controller was trained until reaching the pre-defined maximum amount of training episodes. The learning process can also be manually cut to save computational time. The training phase could be stopped if one observes that the SAC algorithm fails to learn the task by not reaching many waypoints. On the contrary, it can also be stopped if one judges that the SAC-based controller had already achieved satisfactory results and cannot learn anything more from the training episodes.2.**Choosing the trained models:** During the training phase, the parameters of the neural networks of the SAC were constantly evolving. The values of these parameters were regularly saved and stored in specific files during the learning process. We chose to save these parameters every 100 training episodes until episode number 1500, and then every 250 episodes until episode number 5000 (we fixed the maximum amount of training episodes to 5000). All of these saved parameters formed a set of *trained models*. For example, the parameters saved at episode 1000 are denoted as *model 1000*. Based on the number of successes obtained during the training phase, we chose one or more trained models to compare them with the PID controller.3.**Testing phase:** The chosen trained models were independently compared with a PID controller. Each trained model was run on a distinct testing phase of 1000 episodes, composed of 500 test episodes running the SAC-based controller and 500 test episodes running the PID controller. These test episodes reproduced the same task as during the training phase, but then the parameters of the neural networks could not vary anymore (since the learning process was over). The SAC and the PID controllers have to reach the same set of random target waypoints and are subject to the same time-varying ocean currents to preserve the consistency of this comparison.4.**Analyzing the results:** Once the testing phase is finished, the results were analyzed thanks to different performance metrics. These metrics were computed a posteriori using MATLAB.

Now that the whole process of learning and evaluation is described, we will detail how we implemented the PID- and SAC-based controllers.

#### 4.2.1. Implementation of the PID Algorithm

The UUV simulator proposes several ready-to-use templates of classical controllers: PID controllers, PD controllers with compensation of restoring forces for dynamic positioning, model-free sliding mode controllers based on [83,84], model-based sliding mode controllers, model-based feedback linearization controllers, and singularity-free tracking controllers based on [85]. We used the PID controller for the comparison with the SAC algorithm since it is the most used algorithm in the control theory literature.

The PID controller is defined as a multi-input multi-output (MIMO) system and is able to regulate the six degrees of freedom of the RexROV 2 to reach the waypoint—the surge, the sway, the heave, the roll, the pitch, and the yaw (as defined in Section 4.1.2). It computes an input vector u based on the tracking error $e\left(t\right)=r\left(t\right)-\hat{y}$, with r being the reference (corresponding here to the waypoint) and $\hat{y}$ being a measurement of the output variables of the AUV. This vector is defined as follows:(12)$$u\left(t\right)\;=\;K_{p}\;e\left(t\right)\;+\;K_{i}\;\int_{0}^{t}e\left(\tau \right)d\tau \;+\;K_{d}\;\frac{de(t)}{dt}$$
with
(13)$$e\left(t\right)\;=\;{\left[x_{e},y_{e},z_{e},\phi_{e},\theta_{e},\psi_{e}\right]}^{T}$$
where $K_{p}$, $K_{i}$, and $K_{d}$ are real-value matrices called the *gains*, $\left(x_{e},y_{e},z_{e}\right)$ are the errors between the waypoint position and the AUV position (expressed in meters, in Cartesian coordinates, and in the absolute reference frame of Gazebo), and $\left(\phi_{e},\theta_{e},\psi_{e}\right)$ are the Euler angle errors corresponding to the nummber of angles that the AUV lacks to point towards the waypoint (expressed in radians, in the local reference frame of the AUV).

The PID controller provided by the RexROV 2 package was already tuned and its gains were found using the sequential model-based optimization for general algorithm configuration (SMAC) [86] ( https://github.com/automl/SMAC3 (accessed on 10 March 2022)). $K_{p}$, $K_{i}$ and, $K_{d}$ are diagonal matrices defined with the following diagonal coefficients (here, they are rounded for more clarity):(14)$$\left\{\begin{array}{ccc} \mathrm{diag}\_K_{p} & = & {\left[11994,11994,119934,19460,19460,19460\right]}^{T}\\ \mathrm{diag}\_K_{i} & = & {\left[321,321,321,2097,2097,2097\right]}^{T}\\ \mathrm{diag}\_K_{d} & = & {\left[9077,9077,9077,18881,18881,18881\right]}^{T} \end{array}\right.$$

These inputs u are not sent directly to the AUV thrusters. More specifically, the vector u can be written as $u={\left(f_{x},f_{y},f_{z},\tau_{r},\tau_{p},\tau_{y}\right)}^{T}$, where $\left(f_{x},f_{y},f_{z}\right)$ are forces and $\left(\tau_{r},\tau_{p},\tau_{y}\right)$ are torques. The forces and torques provided by the PID controller need to be applied to the AUV in its local reference frame. The UUV simulator uses an intermediary component called the *thruster manager,* which allows transforming these input or force/torque signals given by the PID controller into thruster commands. In the case of the RexROV 2, six thrusters are operating the AUV. As said earlier, each thruster can affect a combination of several degrees of freedom. As shown in Figure 8, the thruster manager defines a *thruster allocation matrix (TAM)* to transform the PID input vector u into a thruster command vector c, as follows:(15)c(t)=TAM.u(t)

The RexROV 2 package gives the following 6 × 6 **TAM** (here, the coefficients are rounded to three digits for more clarity):(16)$$\left(\begin{array}{cccccc} 0.0 & 0.259 & 0.0 & 0.0 & 0.906 & 0.906\\ 0.999 & 0.0 & 0.383 & 0.383 & 0.423 & -0.423\\ 0.0 & 0.966 & 0.924 & -0.924 & 0.0 & 0.0\\ -0.237 & 0.0 & -0.696 & -0.696 & -0.102 & 0.102\\ 0.0 & 0.946 & -0.799 & 0.799 & 0.218 & 0.218\\ 0.488 & 0.0 & 0.331 & 0.331 & -0.764 & 0.764 \end{array}\right)$$

Each line of the **TAM** corresponds to a different thruster of the RexROV 2. After this computation, each component of the command vector c is sent to its respective thruster, following the same indexing as shown in Figure 5. The thruster manager also allows changing specific parameters, such as the maximum thrust, which can be applied to the AUV, the update rate, or the type of thruster.

The guidance algorithm used with the PID controller will simply consist of straight lines going from the initial position of the RexROV 2 to the target waypoints. No navigation algorithm will be used. The variables will be sent directly to the PID by the UUV simulator, with added noises (described in Section 4.1.3).

#### 4.2.2. Implementation of the Soft Actor–Critic Algorithm for AUV Control

The deep reinforcement learning (deep RL) algorithm we chose for controlling the RexROV 2 is the soft actor–critic (SAC), which was previously detailed in Section 3.2.2. The SAC algorithm is one of the latest advances in deep RL. Similar to many deep RL techniques, it is appropriate for continuous control tasks thanks to the use of neural networks, which allows working in continuous state and action spaces (as with mobile robotics).

Policy gradient-based models, such as SAC, are particularly sensitive to the shape of the reward function. The smallest variations can lead to the convergence and not of the model, and the magnitude and the evolution of each term composing the reward function must be chosen carefully. This function is often crafted by trial and error since no global rules exist in the reinforcement learning theory, and the expression of the function can drastically change from one task to another. The simulated environment used during the training must be realistic enough, while not being too complex for the agent. Indeed, if the task requested is too complex, the agent might not be able to converge, e.g., if the state returned by the environment is not descriptive enough, or if the reward function is too constraining. Thus, a trade-off between the reward function and the complexity of the environment must be found.

Since our study involves MIMO dynamical systems, the state $s_{t}$ and the $a_{t}$ will be vectorial from now. Unlike the PID controller, the SAC does not use a thruster manager. It outputs an action vector $A_{t}={\left[u_{1},u_{2},u_{3},u_{4},u_{5},u_{6}\right]}^{T}$, which is composed of the inputs $u_{i}$ sent directly to the six thrusters of the RexROV 2. Each input belongs to the range [−240.0,240.0].

At the time step *t* of a given episode, the environment sends the following state vector $S_{t}$ to the SAC algorithm, composed of 23 variables:(17)$$S_{t}={\left[\;x,\;\Theta ,\;v,\;\omega ,\;\theta_{e},\;\psi_{e},\;x_{e},\;u_{t-1}\;\right]}^{T}$$
where x=(x,y,z) is the position vector of the RexROV 2, Θ=(ϕ,θ,ψ) is its orientation vector (the Euler angles), v is the linear velocity vector (the derivative of x), ω is the angular velocity vector (the derivative of Θ), $\theta_{e}$, and $\psi_{e}$ are, respectively, the tracking errors of the pitch and the yaw angles (the number of angles lacking to make the AUV point towards the waypoint), $x_{e}$ is the error between the position vector x and the position vector of the waypoint, and $u_{t-1}=\left(u_{1},u_{2},u_{3},u_{4},u_{5},u_{6}\right)$ is the action vector of the inputs sent at the previous time step.

To perform the waypoint tracking mission, we designed the following reward function $r_{t}$:(18)$$r_{t}=\left\{\begin{array}{cc} 500 & \mathrm{if}\;\;d_{t}<3\;\;\left(\mathrm{success}\right)\\ -550 & \mathrm{if}\;\;x\;\;\mathrm{or}\;\;y\notin [-20,\;-20]\;\;\mathrm{or}\;\;z\notin [-60,\;-1]\;\;(\mathrm{collision})\\ 40.\exp\left(-\frac{d_{t}}{20}\right) & \mathrm{if}\;\;d_{t}<d_{t-1}\\ -10 & \mathrm{if}\;\;d_{t}\geq d_{t-1} \end{array}\right.$$
where $r_{t}$ is the reward received by the agent at time *t* and $d_{t}$ is the current relative distance between the AUV position and the position of the waypoint to reach.

Each term appearing in (Equation 18) represents a specific feature of the global desired behavior of the AUV. If the AUV enters inside a sphere of 3 m around the waypoint, the agent obtains a reward of 500. If the AUV gets out of the 3D box defined in Section 4.1.3, it obtains a reward of −550. This negative reward has a greater magnitude than when the AUV reaches the waypoint because we want to emphasize the fact that we do not want the AUV to be lost in the environment (it corresponds to the worst-case scenario). We chose to use the exponential term to give greater rewards when the AUV was close to the waypoint. When the AUV moves towards the waypoints ($d_{t}<d_{t-1}$), the more the AUV is close to the waypoint, the greater the rewards (independently of the speed with which the AUV goes towards its goal). When the AUV moves backward to the waypoint or stays still ($d_{t}\geq d_{t-1}$), a negative reward of −10 is systematically given to the AUV, in order to indicate that it is bad behavior.

The four neural networks of the SAC algorithm (detailed in Section 3.2.2) are implemented as follows:**The soft state-value network $V_{\psi }$:** The input of the network has a size of 23 and is composed of the state $S_{t}$ defined in (Equation 17), while its output is the state-value function $V(S_{t})$. It is composed of two hidden layers of 256 neurons and one output layer composed of 1 neuron (corresponding to the estimation of the scalar $V(S_{t})$). The activation function used in the two hidden layers is the leaky rectified linear unit (ReLU), defined as f(x)=max(0.01x,x). The output layer has no activation function and remains a linear layer to perform the regression of the state-value function.**The target network of the soft state-value network**: Its architecture is the same as the soft state-value network $V_{\psi }$.**The soft Q-value network $Q_{w}$**: The input of the network has a size of 29 and is composed of the state $S_{t}$ and the action $A_{t}$, while its output is the Q-value function $Q(S_{t},A_{t})$. It is composed of two hidden layers of 256 neurons and one output layer composed of 1 neuron (corresponding to the estimation of the scalar $Q(S_{t},A_{t})$). The activation function used in the two hidden layers is the leaky ReLU. The output layer has no activation function and remains a linear layer to perform the regression of the action-value function.**The policy network $\pi_{\theta }$**: The input of the network has a size of 23 and is composed of the state $S_{t}$ defined in (Equation 17), while its output is composed of the mean vector $\mu_{\theta }$ and the variance $\sigma_{\theta }$. Since the policy of the SAC algorithm is stochastic, these vectors allow computing the action vector $A_{t}$ using a *squashed Gaussian distribution*:
(19)$$A_{t}=\tan h\left(n\right)\;\;\;\;\mathrm{where}\;n∼N\left(\mu_{\theta },\sigma_{\theta }\right)$$This neural network is composed of two hidden layers of 256 neurons and one output layer composed of 12 neurons: 6 neurons for the components of the vector $\mu_{\theta }$ and 6 neurons for the components of the vector $\sigma_{\theta }$. The activation function used in the two hidden layers is the leaky ReLU. The output layer has no activation function and remains a linear layer to perform the regression of the two mean and variance vectors.

All the components of the weight matrices and the bias vectors of these neural networks were randomly initialized using a uniform distribution over the range $\left[-3\times 10^{-3},\;3\times 10^{-3}\right]$, except for the target network. Its parameters were initialized using a copy of the initial parameters of the soft state-value network.

The parameters of the target network were updated using the rule:$$\psi^{\prime }\leftarrow \tau \psi +\left(1-\tau \right)\psi^{\prime },$$
where τ is set to $5\times 10^{-3}$. The three other neural networks were all updated thanks to the ADAM algorithm [87] and the loss functions defined in Section 3.2.2. All of the learning rates were set to $3\times 10^{-4}$. The updates were computed using the batch of training samples taken from a replay buffer *D* able to contain up to $5\times 10^{6}$ transitions. The size of the batches was set to 256. The temperature parameter α was set to the inverse of the reward range. In this task, we find empirically that the reward function lays around the interval [−20, 20], which means a range of 40. We set the temperature to $\alpha =\frac{1}{40}=0.025$. The discount factor γ was set to 0.99. Many of the previous hyperparameters were set to the values recommended by [67], the original paper on the SAC algorithm. The authors continued to use some of these values in many different use cases.

The SAC algorithm has to perform an end-to-end control of the AUV. This means that it has to carry out both the low-level and high-level control (called guidance) of the AUV. It has to simultaneously send the good inputs to the thrusters of the AUV (low-level) and choose the right trajectories to follow (high-level).

As we focused on the low-level control and the guidance of the RexROV 2, and not on its navigation, the tracking errors of the variables were directly given inside the state vector $S_{t}$. Since the SAC algorithm does not use a thruster manager and it sends its actions directly as the inputs of the thrusters, the task was harder for this deep RL agent. Indeed, it had to figure out the dynamics of the RexROV 2 and how each thruster affected the different degrees of freedom of the AUV.

#### 4.2.3. Definition of the Performance Metrics

When the testing phase was completed, the records of the test episodes of both controllers were analyzed using *MATLAB*. We defined the metrics to measure different aspects of the performance of these controllers, such as the speed of convergence of the learning algorithms, the efficiency in the tracking of the waypoints, and even the energy consumption of the AUV.

The only metric used during the training phase of the SAC algorithm was called the *Number of successes for every 100 episodes*. It counted the number of successful episodes obtained (each one hundred training episodes). It counted the episodes in which the AUV reached the waypoint. It allowed measuring how fast the SAC algorithm was able to reach an acceptable behavior during its learning of the neural network parameters. This metric is presented with the following format: [5, 10, …, 95, 100, 99]. We said earlier that the training phases are composed of a maximum number of 5000 episodes, so the array of the *Number of successes for every 100 episodes* can be composed of 50 numbers or less.

During the testing phase, more metrics were defined to compare the PID and SAC controllers according to different criteria. Here is the list of metrics, all computed on data from the test episodes of both controllers (500 episodes for each of them).

**Success rate:** The success rate is the percentage of successful episodes a controller had throughout its 500 test episodes. As defined in Section 4.1.3, a *success* happened when the AUV reached the waypoint without going outside of the vertical boundaries and without exceeding the maximum number of time steps.**Collision rate:** The collision rate is the percentage of unsuccessful episodes a controller had throughout its 500 test episodes because of collisions. As defined in Section 4.1.3, a *collision* happened when the AUV went beyond the vertical boundaries [−60, −1] on the *z* axis of the environment and the episode ended with a failure.**Timeout failure rate:** The timeout failure rate is the percentage of unsuccessful episodes a controller had throughout its 500 test episodes because of timeout failures. As defined in Section 4.1.3, a *timeout* happened when the AUV spent more than 1000 time steps without reaching the target waypoint and the episode ended with a failure.**Mean of dδ**: We define the notion of *ideal trajectory* as follows: The ideal trajectory is the perfect path allowing the AUV to go directly to the waypoint. In practice, it corresponds to the straight line linking the initial position of the AUV and the target waypoint. We also define the *distance error*
dδ as the measure of the deviation of the AUV from the ideal trajectory. This deviation is expressed in meters and is measured using a 3D line perpendicular to the ideal trajectory and passing through the current position of the AUV (see Figure 9). The *Mean of dδ* is the mean value of the distance error dδ, computed on all the time steps of all the test episodes.**SD of dδ**: By keeping the previous definitions of the ideal trajectory and the distance error dδ, *SD of dδ* is the standard deviation of the distance error dδ, computed on all the time steps of all the test episodes. The mean and the SD of dδ allow assessing the tracking abilities of the controller and are both expressed in meters.**Mean of ∥u∥:** As defined earlier, the vector u is composed of the six commands $u_{i}$ received by the thrusters of the RexROV 2. Its Euclidean norm is defined as $∥u∥=\sqrt{\left({u_{1}}^{2}+{u_{2}}^{2}+{u_{3}}^{2}+{u_{4}}^{2}+{u_{5}}^{2}+{u_{6}}^{2}\right)}$. The *Mean of ∥u∥* is the mean value of the norm ∥u∥, computed on all the time steps of all the test episodes. The mean of ∥u∥ gives an idea of the global thrusters usage made by the controller. The bigger the commands $u_{i}$ are, the more the thrusters will be used. This metric gives (indirectly) information about the durability of an AUV controlled by a given controller: the smaller the demand on the thrusters, the more the AUV actuators will last.**Mean number of steps:** The *mean number of steps* is a metric measuring the mean number of time steps taken by a test episode of a given controller. It simply gives the mean duration of the episodes performed during the testing process, independently of the results of these episodes (success or failure). The mean number of steps is correlated to the mean and the SD of dδ since a great number of time steps taken by an episode can be due to the fact that the AUV deviates too much from the ideal trajectory, and that it has struggled to reach the target waypoint.**Mean of ∑∥u∥:** For each episode, the sum of the norm ∥u∥ is computed, noted as ∑∥u∥. It gives a precise idea of the total amplitude of all the commands asked to the thrusters during each episode. The *Mean of ∑∥u∥* is simply the mean value of the quantity ∑∥u∥ computed on all the test episodes. These metrics give the mean thrusters usage per episode and give (indirectly) information about the energy consumption of the AUV. The greater the quantity ∑∥u∥ is for one episode, the bigger the energy consumption of the AUV is during that episode. Indeed, the greater the command $u_{i}$ is, the greater the electric current sent to the thruster *i* will be. The values of the mean of ∑∥u∥ shown in the results of the following sections will always have to be multiplied by $10^{5}$ to have the true numbers. We multiplied the values of the mean of ∑∥u∥ by $10^{-5}$ to not have too large numbers in our results tables.

The results of the test episodes can either be a success or a failure and can bias the metrics based on the distance error dδ, the time steps, and the norm ∥u∥, depending on what we wanted to analyze. All of the cited metrics were computed using only successful episodes, and these additional values are denoted with the term *(Success)* in front of their names. For example, these *(Success)* versions of the metrics can be employed to know the mean time needed by the controllers to reach the waypoints, or the amplitudes of the deviations from the ideal trajectory caused only by the ocean currents (and not by the aberrant behavior of a machine learning-based controller that has not learned the task well).

Finally, all the metrics will not be systematically commented on. We display tables giving the maximum amount of information to the reader, but we only discuss the most relevant values for each trial. Moreover, some metrics were also implicitly included in one another and can be considered intermediary values. For example, the mean of ∑∥u∥ can be viewed as an approximation of the product of the mean of ∥u∥ and the mean number of steps. It is the expression of an energy value and the product of a time value and an intensity value.

## 5. Results

In this section, we describe the results of the comparison of the SAC with the PID controller on the waypoint tracking task of the RexROV 2. Moreover, we studied the impact of the state vector components on the performance of the SAC in this task. We conducted many different trials, during which, we changed the composition of the stat vector given by the environment to the agent.

We started with the same state vector $S_{t}$ (the vector defined in Section 4.2.2), and we successively removed variables until the agent was no more capable of learning the task. We wanted to know the maximum number of variables we could remove from the state vector while still being able to fulfill the control task. The task became harder for the agent when it obtained less information. If the SAC was able to learn the task with fewer variables, this meant that some sensors could be removed from the RexROV 2, and it allowed identifying which sensors were the more useful for understanding the AUV dynamics.

In this section, we display a selection of all the trials we performed (only the most interesting trials are shown). All of the training and testing simulations were performed in real-time.

### 5.1. Initial State Vector

We first trained the SAC algorithm with the new changes described previously but with the same state vector $S_{t}$ as for the task of Section 4.2.2:(20)$$S_{t}={\left[\;x,\;\Theta ,\;v,\;\omega ,\;\theta_{e},\;\psi_{e},\;x_{e},\;u_{t-1}\;\right]}^{T}$$

Let us recall that x=(x,y,z) is the position vector of the RexROV 2, Θ=(ϕ,θ,ψ) is its orientation vector (the Euler angles), v is the linear velocity vector, ω is the angular velocity vector, $\theta_{e}$, and $\psi_{e}$ are the tracking errors of the pitch and the yaw angles, respectively, $x_{e}$ is the error between the position vector x and the position vector of the waypoint, and $u_{t-1}=\left(u_{1},u_{2},u_{3},u_{4},u_{5},u_{6}\right)$ is the action vector of the inputs sent at the previous time step. This vector is composed of 23 dimensions.

We trained the SAC algorithm for 2700 training episodes, and we had the following number of successes.

**Number of successes for every 100 episodes:** [12, 60, 85, 95, 99, 99, 98, 99, 98, 100, 100, 100, 100, 100, 99, 100, 100, 100, 99, 99, 100, 100, 100, 100, 99, 100, 100]

The number of episodes after which we stopped the training phase varied during the following trials since there was no rule of thumb for the number of training episodes needed for this task. Here, the SAC managed to converge rapidly towards good behavior. It reached an 85% success rate after only 300 episodes. This shows that the SAC algorithm is sample efficient, which means that it needs a few episodes to learn this control task.

After the training process, we tested three different models to see how the number of training episodes performed by the agent could affect its performance during the testing phase. We tested a model after 600 training episodes, a model after 1300 training episodes, and a model after 2500 training episodes. These three models of the SAC algorithm were compared with the PID controller during 1000 episodes, which corresponds to 500 testing episodes for each controller. Here are the three results (Table 1, Table 2 and Table 3).

These tables display satisfying results. For each metric, we highlighted the controllers with the best results (highlighted in green). First, **the primary goal of the task was fulfilled:** for the SAC to have a success rate superior or equal to the PID controller. On this run, model 600 had the same success rate as the PID (within 0.2%), and models 1300 and 2500 performed slightly better (1% and 1.2% better, respectively). Moreover, even if the PID remained closer to the ideal trajectory (the mean and SD of dδ of the PID controller were always lower than of the SAC), the mean of ∑∥u∥ was always lower than the PID. The SAC managed to save more energy than the PID controller, which was our secondary objective for this task. These observations stayed true when we took into account all episodes, but also with only the successful episodes (it corresponded to the metrics with the tag *Success*). The models 1300 and 2500 also had slightly fewer collision failures and took smaller numbers of time steps to finish their test episodes, which was always good (even if it was not the focus of this work).

The SAC performed as well as the PID controller in terms of success, and even better for some models. However, we did not expect to have better energy consumption while having such success rates. **This means that the SAC algorithm was able to find a good trade-off between fulfilling the control task (reaching waypoints) and saving energy (generating smaller cumulative inputs during each episode)**.

### 5.2. Removing the Measure of the Position Vector, the Euler Angles, and the Pitch Tracking Error

After a successful trial where we only removed the position vector x=(x,y,z) of the AUV from the state vector of the SAC (this trial is not detailed here), we then removed the measure of the Euler angles, given by the orientation vector Θ=(ϕ,θ,ψ), and the pitch tracking error $\theta_{e}$ (the number of pitch angles lacking to make the AUV point towards the waypoint). $S_{t}$ became a 16-dimensional vector:(21)$$S_{t}={\left[\;v,\;\omega ,\;\psi_{e},\;x_{e},\;u_{t-1}\;\right]}^{T}$$

After 3400 training episodes, the SAC had the following number of successes during the training phase:
**Number of successes for every 100 episodes:** [43, 100, 98, 100, 100, 100, 100, 100, 100, 100, 100, 100, 100, 100, 99, 99, 100, 100, 100, 99, 100, 100, 100, 100, 99, 100, 100, 99, 100, 99, 100, 100, 100, 100]

The SAC algorithm was faster at converging towards good behavior than during the trial of Section 5.1. It had more than 40% during the first 100 episodes, and it managed to reach a 100% success rate after only 200 episodes. **The SAC algorithm had good learning abilities on this control task since the agent managed to learn the task faster than during the previous trial, but with less information given from the environment**.

The models selected after 600, 1300, and 2500 training episodes were tested and compared with the PID controller. Here are the results of these three test phases (Table 4, Table 5 and Table 6).

The three models had better success and collision rates than the PID controller, and **the model trained on 1300 episodes even achieved a success rate of 99.4%, meaning that it did not reach the waypoint during the three test episodes**. The PID controller had a better mean and a better SD for the distance error dδ. The models 600 and 1300 managed to save more energy than the PID since they had a lower mean ∑∥u∥, but for the first time, the PID managed to beat model 2500 on this criterion (on all episodes and successful episodes).

These results show that **the SAC algorithm does not need to know its orientation vector Θ or the pitch tracking error $\theta_{e}$ to perform the task. The only angle it needs to know is the yaw tracking error $\psi_{e}$. It can also control the RexROV 2 without knowing its true global position vector. However, it still needs to know its position relative to the waypoint, given by the error position vector $x_{e}$**.

### 5.3. Removing the Measure of the Angular and the Linear Speeds

After a successful trial where we only removed the angular velocity vector ω (this trial is not detailed here), we removed the linear velocity vector v. The vector $S_{t}$ now has 10 dimensions:(22)$$S_{t}={\left[\;\psi_{e},\;x_{e},\;u_{t-1}\;\right]}^{T}$$

After 2500 training episodes, the SAC had the following number of successes during the training phase:
**Number of successes for every 100 episodes:** [1, 26, 48, 76, 78, 87, 89, 90, 72, 77, 81, 85, 89, 90, 91, 95, 96, 96, 62, 68, 24, 47, 45, 61, 46]

The SAC never reached a success rate of 100% during this training phase, but it managed to have more than a 95% success rate several times.

Since the training phase was less successful than during the previous trials, we only tested one model. We chose to test model 1750 by examining the success rates obtained during the learning process. The SAC had a 96% success rate in 200 successive episodes, between episodes 1601 and 1800. Here are the corresponding results (Table 7).

Model 1750 reached a 97.6% success rate. Even if the PID performed slightly better (with a difference of 1.6% between the two success rates), it was still an excellent performance from the SAC algorithm. The success rate did not reach the symbolic 100% barrier during this training phase, but the task was performed just as well as in the previous testing phases. **The PID controller saved more energy than the SAC on all episodes but it was the contrary if we took into account only the successful episodes**. For us, the mean of ∑∥u∥ computed on the successful episodes was more meaningful than when it was computed on all of the episodes. It allowed us to better compare the tracking abilities of the controllers, without being biased by the episodes where the AUV went outside of the box, or stagnated without reaching the target waypoint. We can also note that **the SAC had surprisingly better success rates during the testing phase than during the training phase**, where it reached 96% at most.

**The SAC algorithm was still able to learn the waypoint tracking task and understand the AUV dynamics without any velocity information (angular or linear)**.

### 5.4. Removing the Values of the Previous Inputs

For this last trial, we removed the vector of the past inputs $u_{t-1}$ from $S_{t}$, leaving the state vector with only four dimensions:(23)$$S_{t}={\left[\;\psi_{e},\;x_{e}\;\right]}^{T}$$

After 2500 training episodes, the SAC had the following number of successes during the training phase:
**Number of successes for every 100 episodes:** [6, 16, 45, 67, 65, 70, 70, 65, 68, 68, 71, 77, 52, 53, 54, 58, 45, 45, 57, 36, 53, 73, 63, 63, 46].

The learning process was not more than 80%, so we cannot say that the SAC agent converged towards satisfactory behavior. **However, it still reached more than 75% and it is still worth it to compare it with the PID controller. We can assume that this state vector configuration made the agent struggle during the training phase**.

Similar to the previous trial, we only tested model 1100. During the training phase, the success rate reached 71% before episode 1100 and 77% afterward. Here is the corresponding results table (Table 8).

Compared to the results of Section 5.3 (where the SAC obtained a success rate of 97.6%), **removing the past inputs from the state vector made the success drop to 75.4%**. The collision and timeout failure rates strongly increased, which further confirmed the drop in performance. Moreover, **the SAC-based controller was not able to save more energy than the PID controller, making it worst than the PID in all of the criteria**.

Even if the PID performed better in this trial, the SAC algorithm managed to have sub-optimal behavior with a state vector of only four dimensions. A success rate of 75.4% still showed that the SAC algorithm was able to learn the task and understand the dynamics of the RexROV 2 with a minimal amount of information. Its performance was just not good enough to challenge the PID controller.

## 6. Discussion

In this section, we highlight global observations about our results and discuss the benefits provided by our approach. Many observations have already been presented in the previous sections; here, we only summarize the main results.

### 6.1. Summary of the Results

Thanks to the sensitivity analysis we made on the size of the state vector, the SAC algorithm managed to fulfill this waypoint tracking task with a reduced state vector $S_{t}$. **In Section 5.3, the state vector of the SAC algorithm was set to its smallest size while still allowing the controller to perform as well as the PID controller**. The state vector was reduced from 23 to 10 dimensions and was composed of the following variables:(24)$$S_{t}={\left[\;\psi_{e},\;x_{e},\;u_{t-1}\;\right]}^{T}$$

In almost all cases until this configuration, the SAC algorithm managed to equalize the success rate of the PID controller, and often it had a slightly better one. Even if the PID controller was always closer to the ideal trajectory, the SAC almost always saved more energy than the PID by generating lower cumulative inputs during the test episodes. It could allow performing longer missions.

**The best success rate was obtained by model 1300 tested in Section 5.2**. The state vector of the SAC algorithm was then composed of 16 dimensions:(25)$$S_{t}={\left[\;v,\;\omega ,\;\psi_{e},\;x_{e},\;u_{t-1}\;\right]}^{T}$$

**The best energy-saving performance** was assigned to the model achieving the lowest mean of ∑∥u∥. It corresponded to model 2500 of Section 5.1, trained with the initial 23-dimensional state vector:(26)$$S_{t}={\left[\;x,\;\Theta ,\;v,\;\omega ,\;\theta_{e},\;\psi_{e},\;x_{e},\;u_{t-1}\;\right]}^{T}$$

Based on all of these results, we can also note that **the number of training episodes did not influence the performance after a certain threshold**. Once the SAC algorithm converged towards satisfactory behavior, training on more episodes did not improve the success rates of the SAC-based controller. This threshold can be found in the array of the *Number of successes for every 100 episodes* metric. Once the training reached more than 70% of success several times in a row, we considered the model sufficiently trained. If the performance of a given model was not suitable or if we wanted to see that a model could have better results, we could try another model by selecting it from the ones beyond the threshold. This process of selecting the right model is rather empirical and task-dependent.

### 6.2. Our Main Contributions

In this section, we list the main contributions provided by our results.

#### 6.2.1. Making the SAC Control the RexROV 2 on a Waypoint Tracking Task

First, our main objective was to analyze if the SAC algorithm was able to understand the dynamics of an AUV and to learn how to perform a waypoint tracking task while dealing with the varying ocean currents and noises found in the sensors and the actuators of the RexROV 2. **In all of our tests, we managed to make the SAC successfully converge towards correct behavior**. During the testing phases, it reached a success rate of more than 90% for many trials (except in Section 5.4). We found a reward function that was able to match the needs of the control task and the AUV.

Moreover, **the SAC was able to understand the dynamics induced by the particular layout of the AUV thrusters (described in Section 4.1.2), without the need for a thruster manager and a thruster allocation matrix (TAM)**, such as the implementation of the PID controller found in the UUV simulator (see Section 4.2.1). This demonstrates the effectiveness of the learning abilities of the SAC algorithm.

As explained in Section 4.2.2, **the SAC algorithm also replaced both the control and the guidance algorithms**. It was able to generate both the trajectories, allowing us to reach the waypoint (guidance) and the inputs needed to follow these trajectories (control). Figure 10 shows the scope of the SAC-based controller. **This algorithm alone allows replacing the guidance algorithm, the PID controller, and the TAM**.

Another benefit of the use of the SAC algorithm is that it is able to control the six degrees of freedom (DOF) of the AUV simultaneously. On the contrary, the PID controller controls each DOF independently. **Being able to control the 6DOF simultaneously is a real advantage since it allows one to better take into account the coupling found between the DOF and the nonlinearities of the AUV**.

The SAC algorithm is a model-free algorithm, which means that no AUV model (e.g., Fossen’s model [79]) is needed during the design of this AUV controller. This is a great feature since **this controller could be applied to other types of dynamical systems (not only to AUVs)**. It will only require the tuning of some of the hyperparameters.

Finally, the waypoint tracking task is a general task. Since this SAC-based controller is able to reach target waypoints, **it can be applied to any control tasks involving waypoint tracking or path-following** since a given trajectory can be decomposed in multiple waypoints.

#### 6.2.2. Challenging the PID Controller

During our trials, the SAC-based controller was compared with PID controllers. Several performance metrics were used in these testing phases to measure various aspects of the performance, e.g., the percentage of success computed from the number of reached waypoints, the power consumption, the deviation from the expected paths, etc.

In almost all testing phases, the SAC algorithm managed to equalize and even sometimes surpass the PID controller in terms of the success rate. Moreover, even if the SAC-based controller took systematically more time to reach the waypoints than the PID, it almost always consumed less energy. This was due to the fact that the SAC generated average thruster input signals of smaller magnitudes. For a given test episode, the sum of all the inputs sent by the SAC to the AUV remained smaller than the PID controller inputs.

Generally, the SAC algorithm deviated more from the expected path (what we named the *ideal trajectory*) than the PID controller. These deviations (measured in our results by the *distance error* dδ metric) were due to the sudden variations found in both the magnitudes and the directions of the ocean currents. The SAC was able to adapt its behavior to overcome these unexpected external disturbances and still reach the waypoints despite these large deviations. Moreover, the SAC-based controller was able to control the AUV despite the noises added to the sensors and actuator variables. All of these disturbances are detailed in Section 4.1.3.

Even if the target waypoints took more time steps before being reached by the AUV, the **SAC managed to outperform the PID controller in terms of power consumption, while still having (at least) a similar success rate. It was able to find a trade-off between performance and consumption**.

#### 6.2.3. Reducing the Size of the State Vector

Another goal of our work was to analyze how the composition of the state vector $S_{t}$ given to the SAC agent could affect the performance of the deep reinforcement learning-based controller. We began our trials with an initial 23-dimensional state vector. This vector included typical AUV sensor measurements (measuring the variables described in Section 4.1.2), as well as information about the tracking errors with respect to the target waypoints (usually given by a guidance algorithm). We managed to progressively reduce the size of the state vector by removing variables (in trial after trial), and the SAC-based controller could still fulfill the waypoint tracking task in most cases. Among the configurations able to challenge the PID controller, the lowest successful configuration was reached by setting the state vector to 10 variables (Section 5.3). In this configuration, the controller obtained a success rate of 97.6% with less information, which is almost a perfect score. **Having a state vector with fewer variables means that fewer sensor measurements are needed by the agent. Therefore, we can remove sensors from the AUV, which will reduce the production costs of the robot**.

As described in Section 4.1.2, the initial sensors embedded inside the RexROV 2 were the inertial measurement unit (IMU), a Doppler velocity log (DVL), and a pressure sensor. Moreover, the target waypoint must have either been known or sent its location using acoustics communication. Thus, the RexROV 2 must also include acoustics transducers to create long-baseline (LBL), short baseline (SBL), or ultra-short baseline (USBL) acoustic positioning systems [88]. A GPS intelligent buoys (GIB) system is also possible [89].

In the case of the state vector configuration found in Section 5.3, the transducers were kept, but the IMU, DVL, and pressure sensor could have been removed. Instead, we only needed a compass to know the heading of the AUV. With the transducers and the compass, we can know the position error $x_{e}$ and the yaw error $\psi_{e}$. The previous inputs $u_{t-1}$ are simply recorded over time.

## 7. Conclusions and Openings

The sensitivity analysis of the state vector presented in this work allowed analyzing which sensors are mandatory to allow the SAC algorithm to control the RexROV 2. After having successfully challenged the PID controller inside marine robotics simulations, our next objective will be to embed our SAC-based controller in a real-world AUV.

We had this next step in mind since the beginning of this work and we implemented choices allowing to facilitate the transfer towards real-world robots.

First, we coded our controllers using ROS middleware (mentioned in Section 4.1.1), which is known for providing great flexibility and allowing to easily transfer its software components (called *nodes*) towards embedded systems. Its component-based architecture also allows easily changing the type of robot, sensors, actuators, as well as the algorithms used in the GNC system. We will only have to change several hyperparameters to make the transfer from the simulation to the real world, even if the real AUV is not the same as the RexROV 2.

Moreover, we set our simulations to be computed in real-time. After many observations, we noticed that changing the speed at which the simulation was computed could change the behavior of the SAC algorithm. This was due to the fact that ROS executed its nodes in an asynchronous way. When the simulation is computed at a different speed, the SAC (executed in an independent node) samples the environment at a different rate than before the change, and its actions are not maintained during the same number of time steps.

This means that a DRL-based controller, which obtained good results during the simulated testing phases, will not necessarily perform the same when it is embedded in a real robotic platform. From the point of view of the agent of the DRL algorithm, the environment does not act the same as during its training phase.

Therefore, executing the simulation in real-time guarantees that the behavior of the SAC-based controller will be the same in a real-world robot as during our simulation results.

In terms of architecture, the three neural networks (NNs) implemented by the SAC algorithm (detailed in Section 4.2.2) are shallow networks. They are all composed of only two hidden layers of 256 neurons. These are light NNs in terms of the number of parameters needed to be updated during the learning process.

Moreover, the smaller the state vector is (as in our results), the smaller the input layer of these NNs will be. This means that less memory usage and computational resources will be needed.

These two elements allow the SAC algorithm to be embedded inside platforms with less memory (RAM) and computational (CPU) resources. Thus, this DRL-based controller can be applied to a broader range of robots. For an example of the memory usage used by this DRL-based controller, the SAC algorithm used 3.6 GB of RAM and 600 MB of V-RAM (the RAM dedicated to the GPU).

We included in our pipeline more ideas taken from works about the sim-to-real transfer of deep reinforcement learning algorithms [90]. This includes elements from the subfields of domain randomization, domain adaptation, imitation learning, meta reinforcement learning, and knowledge distillation.

For example, the concept behind domain randomization [91] is to highly randomize the simulated environment to cover the real distribution of the real-world data despite the bias between the model world and the real world, instead of having to find the most realistic simulation possible (which can be a hard (and time-consuming) task). In our work, we already randomized the ocean currents and the location of the target waypoints, but we could go further by randomizing the starting point of the AUV and iteratively adding new random elements, such as rocks or other vehicles. We could also successively simulate different types of probability distributions, not only continuous uniform distributions.

Domain adaptation is another interesting subfield belonging to the transfer learning approaches [92]. The idea is to transfer the knowledge between a source domain where many data are available to a target domain where data are lacking. Of course, the source and target domains have to be closely related. The features used in both tasks are extracted from their respective feature spaces and are then used inside a unified feature space to facilitate the transfer. In our example, the source domain would be the simulated environment and the target domain would be the real-world environment. This would allow transferring our controller from the simulation to the real world with a few real-world training episodes, thanks to the domain application.

The fact that sim-to-real techniques have already been successfully tested for the SAC algorithm, for example in [93,94,95], is encouraging for the transfer of our SAC-based controller to a real AUV.

If we look at our work from a long-term perspective—our work falls within the project of facilitating the control of AUVs thanks to the use of ML. The proposals made here represent the first step towards this goal and allow identifying the most important aspects to handle during the implementation of RL algorithms for the end-to-end control of AUVs. We believe that this goal can be achieved thanks to the progressive integration of elements taken from the control theory or robotics inside the ML methods. We prefer to have the best of both worlds—the adaptability and the innovative control approaches achieved by the ML, as well as the guarantees and the knowledge provided by the control theory and robotics. Some works found in the literature developed RL methods based on elements taken from the control theory and show that this unification is possible. The integration of expert knowledge in ML could result in the creation of hybrid methods. Specifically, we prefer to test methods from a safe RL [96,97], to combine safety approaches with the SAC algorithm. These methods could add safety and stability guarantees to the learning process.

The code corresponding to this work is available on GitHub: https://github.com/YoannSOLA/Soft-Actor-Critic_for_RexROV2 (accessed on 20 June 2022).

## Figures and Tables

**Figure 1 sensors-22-06072-f001:**
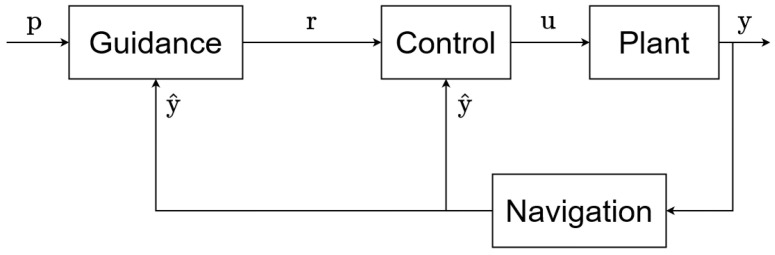
A general guidance–navigation–control (GNC) control.

**Figure 2 sensors-22-06072-f002:**
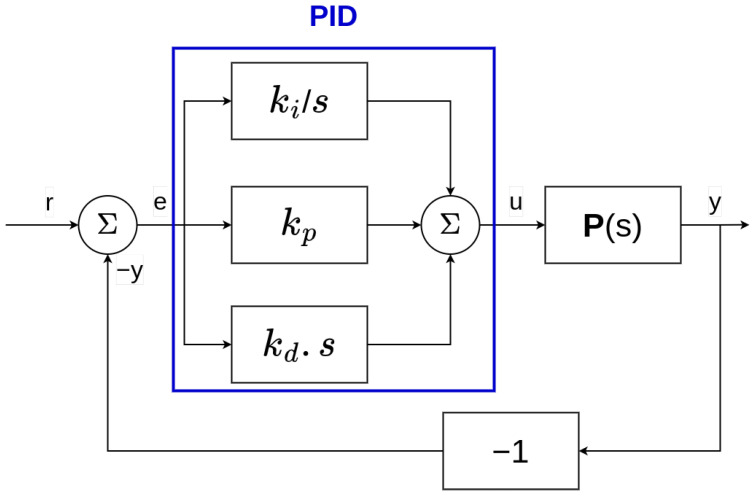
The block diagram of a feedback system composed of a PID controller and a plant model.

**Figure 3 sensors-22-06072-f003:**
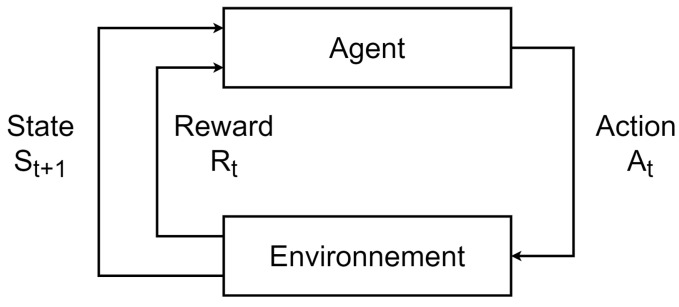
A Markov decision process.

**Figure 4 sensors-22-06072-f004:**
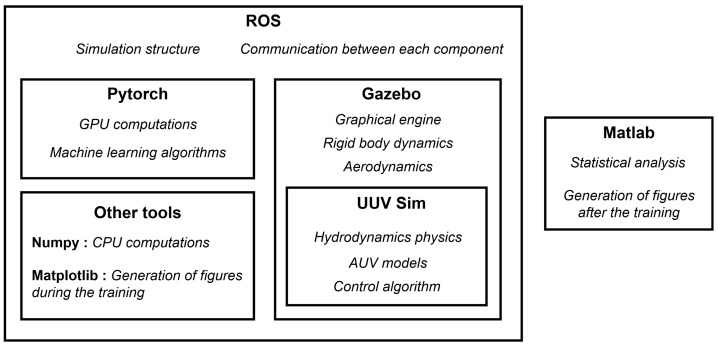
Architecture of our marine robotics simulation.

**Figure 5 sensors-22-06072-f005:**
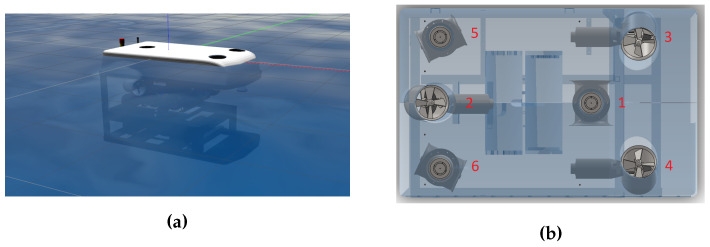
The RexROV 2 model available in the UUV simulator. (**a**) The RexROV 2 shown in the Gazebo client. (**b**) Thruster layout of the RexROV 2 (top view).

**Figure 6 sensors-22-06072-f006:**
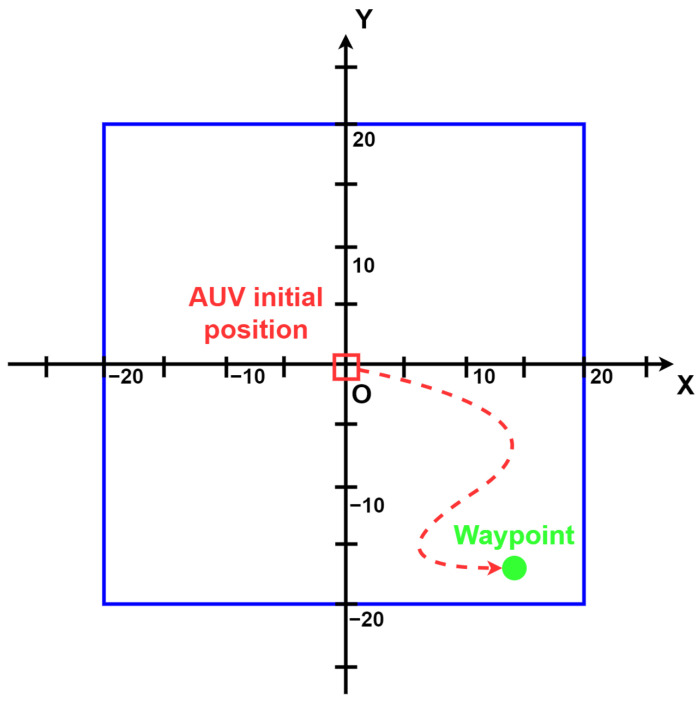
The bounded boxes of the control task represented in the horizontal plane (O,X,Y).

**Figure 7 sensors-22-06072-f007:**
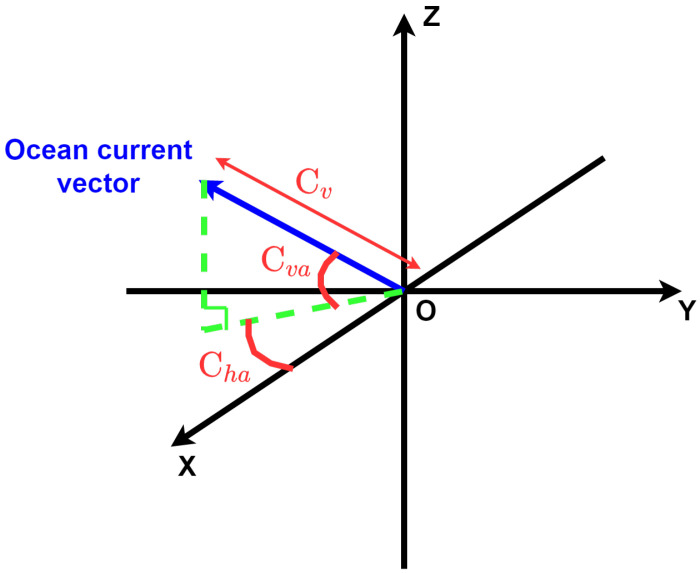
An example of an ocean current vector in the Gazebo reference frame.

**Figure 8 sensors-22-06072-f008:**
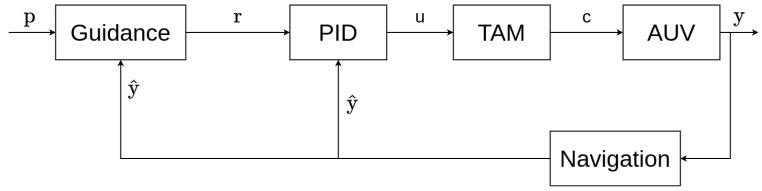
The GNC system with the PID as the controller and the addition of the thruster allocation matrix (TAM).

**Figure 9 sensors-22-06072-f009:**
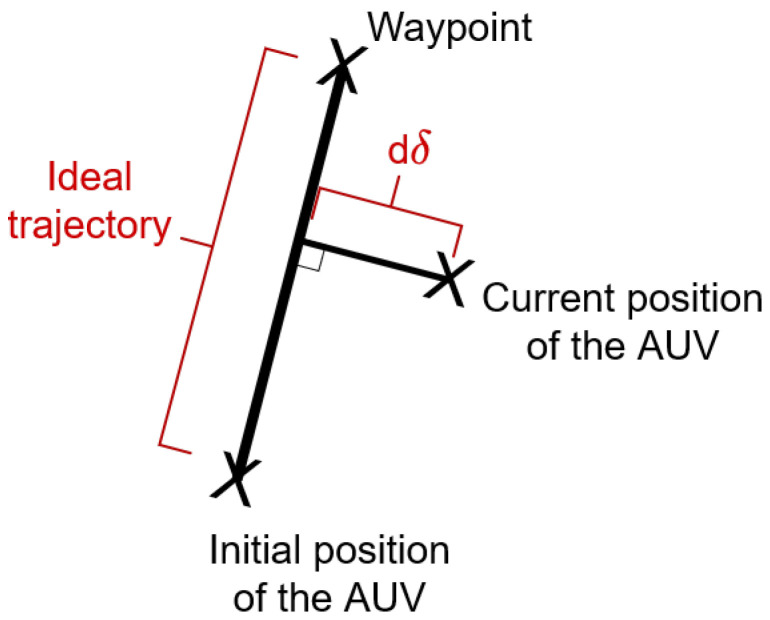
Definition of the ideal trajectory and the distance error dδ.

**Figure 10 sensors-22-06072-f010:**
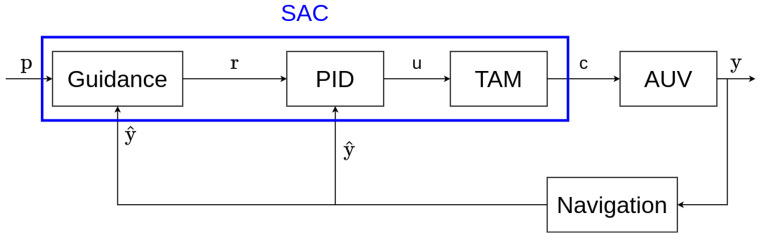
The components of the GNC system being replaced by the SAC-based controller.

**Table 1 sensors-22-06072-t001:** Testing phase of a model based on the initial state vector, trained on 600 episodes. For each metric, the controller with the best score is highlighted in green.

Metrics	PID	SAC
Success rate (%)	97.2	97.0
Collision rate (%)	2.8	3.0
Timeout failure rate (%)	0.0	0.0
Mean of dδ (m)	1.56	3.16
SD of dδ (m)	1.46	1.80
Mean of ∥u∥	531.31	413.36
Mean number of steps	200	209
Mean of ∑∥u∥ $\left(\times 10^{5}\right)$	1.065	0.864
(Success) Mean of dδ (m)	1.42	2.95
(Success) SD of dδ (m)	1.27	1.57
(Success) Mean of ∥u∥	531.39	413.73
(Success) Mean number of steps	204	212
(Success) Mean of ∑∥u∥ $\left(\times 10^{5}\right)$	1.081	0.874

**Table 2 sensors-22-06072-t002:** Testing phase of a model based on the initial state vector, trained on 1300 episodes. For each metric, the controller with the best score is highlighted in green.

Metrics	PID	SAC
Success rate (%)	97.6	98.6
Collision rate (%)	2.2	1.4
Timeout failure rate (%)	0.2	0.0
Mean of dδ (m)	1.50	2.66
SD of dδ (m)	1.35	1.44
Mean of ∥u∥	531.63	446.00
Mean number of steps	202	175
Mean of ∑∥u∥ $\left(\times 10^{5}\right)$	1.074	0.780
(Success) Mean of dδ (m)	1.37	2.64
(Success) SD of dδ (m)	1.17	1.41
(Success) Mean of ∥u∥	531.83	446.06
(Success) Mean number of steps	205	178
(Success) Mean of ∑∥u∥ $\left(\times 10^{5}\right)$	1.085	0.790

**Table 3 sensors-22-06072-t003:** Testing phase of a model based on the initial state vector, trained on 2500 episodes. For each metric, the controller with the best score is highlighted in green.

Metrics	PID	SAC
Success rate (%)	97.0	98.2
Collision rate (%)	3.0	1.6
Timeout failure rate (%)	0.0	0.2
Mean of dδ (m)	1.73	2.95
SD of dδ (m)	1.63	2.11
Mean of ∥u∥	531.70	467.97
Mean number of steps	199	154
Mean of ∑∥u∥ $\left(\times 10^{5}\right)$	1.057	0.720
(Success) Mean of dδ (m)	1.49	2.55
(Success) SD of dδ (m)	1.31	1.59
(Success) Mean of ∥u∥	531.69	469.55
(Success) Mean number of steps	203	155
(Success) Mean of ∑∥u∥ $\left(\times 10^{5}\right)$	1.075	0.722

**Table 4 sensors-22-06072-t004:** Testing phase of a model after removing the position vector, the Euler angles, and the pitch error from the state vector, trained on 600 episodes. For each metric, the controller with the best score is highlighted in green.

Metrics	PID	SAC
Success rate (%)	97.0	97.6
Collision rate (%)	2.8	2.4
Timeout failure rate (%)	0.2	0.0
Mean of dδ (m)	1.85	4.27
SD of dδ (m)	1.76	2.59
Mean of ∥u∥	532.72	429.72
Mean number of steps	209	239
Mean of ∑∥u∥ $\left(\times 10^{5}\right)$	1.113	1.027
(Success) Mean of dδ (m)	1.54	4.21
(Success) SD of dδ (m)	1.35	2.51
(Success) Mean of ∥u∥	532.66	429.52
(Success) Mean number of steps	214	242
(Success) Mean of ∑∥u∥ $\left(\times 10^{5}\right)$	1.134	1.034

**Table 5 sensors-22-06072-t005:** Testing phase of a model after removing the position vector, the Euler angles, and the pitch error from the state vector, trained on 1300 episodes. For each metric, the controller with the best score is highlighted in green.

Metrics	PID	SAC
Success rate (%)	97.8	99.4
Collision rate (%)	2.2	0.6
Timeout failure rate (%)	0.0	0.0
Mean of dδ (m)	1.50	3.16
SD of dδ (m)	1.30	1.70
Mean of ∥u∥	533.04	467.87
Mean number of steps	203	189
Mean of ∑∥u∥ $\left(\times 10^{5}\right)$	1.081	0.883
(Success) Mean of dδ (m)	1.33	3.15
(Success) SD of dδ (m)	1.08	1.69
(Success) Mean of ∥u∥	533.12	467.91
(Success) Mean number of steps	206	190
(Success) Mean of ∑∥u∥ $\left(\times 10^{5}\right)$	1.092	0.887

**Table 6 sensors-22-06072-t006:** Testing phase of a model after removing the position vector, the Euler angles, and the pitch error from the state vector, trained on 2500 episodes. For each metric, the controller with the best score is highlighted in green.

Metrics	PID	SAC
Success rate (%)	96.2	98.8
Collision rate (%)	3.8	1.2
Timeout failure rate (%)	0.0	0.0
Mean of dδ (m)	1.45	4.96
SD of dδ (m)	1.40	2.74
Mean of ∥u∥	532.82	457.09
Mean number of steps	199	336
Mean of ∑∥u∥ $\left(\times 10^{5}\right)$	1.058	1.536
(Success) Mean of dδ (m)	1.30	4.86
(Success) SD of dδ (m)	1.19	2.65
(Success) Mean of ∥u∥	532.82	457.26
(Success) Mean number of steps	206	335
(Success) Mean of ∑∥u∥ $\left(\times 10^{5}\right)$	1.093	1.529

**Table 7 sensors-22-06072-t007:** Testing phase of a model after removing the angular and linear speeds from the state vector, trained on 1750 episodes. For each metric, the controller with the best score is highlighted in green.

Metrics	PID	SAC
Success rate (%)	99.2	97.6
Collision rate (%)	0.2	0.8
Timeout failure rate (%)	0.6	1.6
Mean of dδ (m)	1.63	6.54
SD of dδ (m)	1.39	4.93
Mean of ∥u∥	532.62	459.05
Mean number of steps	208	248
Mean of ∑∥u∥ $\left(\times 10^{5}\right)$	1.109	1.137
(Success) Mean of dδ (m)	1.45	4.84
(Success) SD of dδ (m)	1.20	2.90
(Success) Mean of ∥u∥	532.31	461.29
(Success) Mean number of steps	205	234
(Success) Mean of ∑∥u∥ $\left(\times 10^{5}\right)$	1.085	1.077

**Table 8 sensors-22-06072-t008:** The testing phase of a model after removing the linear speeds from the past inputs, trained on 1100 episodes. For each metric, the controller with the best score is highlighted in green.

Metrics	PID	SAC
Success rate (%)	97.8	75.4
Collision rate (%)	2.0	16.4
Timeout failure rate (%)	0.2	8.2
Mean of dδ (m)	2.42	6.36
SD of dδ (m)	2.53	3.94
Mean of ∥u∥	532.78	427.06
Mean number of steps	202	354
Mean of ∑∥u∥ $\left(\times 10^{5}\right)$	1.075	1.512
(Success) Mean of dδ (m)	1.50	4.81
(Success) SD of dδ (m)	1.23	2.74
(Success) Mean of ∥u∥	532.74	430.42
(Success) Mean number of steps	204	295
(Success) Mean of ∑∥u∥ $\left(\times 10^{5}\right)$	1.082	1.265

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
