# Peer review of "Simultaneous Control and Guidance of an AUV Based on Soft Actor–Critic"

_sensors, 2022, doi:10.3390/s22166072_

Round 1

Reviewer 1 Report

This paper deals with the problem of facilitating the control of AUVs thanks to the use of machine learning (ML). The main contribution is an extensive simulation setting.   I only suggest to improve the Introduction by discussing the relationships with ML approaches proposed for UAV, see, e.g.,   - Machine learning techniques in internet of UAVs for smart cities applications, Journal of Intelligent & Fuzzy Systems, 2022   - Machine Learning Methods for UAV Flocks Management-A Survey, IEEE Access 2021   - On the application of machine learning to the design of UAV-based 5G radio access networks Electronics, 2020

Author Response

Hello, thank you for your review and your comments.
I took into account your suggestion and I added a section called "2. Related works" where I talk about existing works and the relationship between  ML and general robotics, UAV and AUV.
One of the paper you gave as an example is cited inside this part.
This new section goes from line 57 to line 181 (I am referring to the line numbers found on the right, in the PDF generated by latex).

Reviewer 2 Report

The paper studies the application of a reinforcement learning method, namely the soft actor-critic method to the problem of waypoint tracking of an AUV. The authors show that the proposed method is competitive with respect to classic methods based on PID control through realistic simulations with the Gazebo dynamics simulator with the UUV plugin. The paper is well written and clear, the analysis of the results is torough and its topics are very timely. I just have two issues that I would like that the authors to discuss better:

- It is not clear which sensors the vehicles use during simulations. Underwater navigation is a challenging problem and often the estimation of the state of the vehicle may not be very accurate. It is not clear if that factor is taken into account during the simulations.

- I would also like the authors to discuss the applicability of these methods in a real setting. My main concern is that to obtain a good policy there is the need for hundreds of training episodes, and this may not be feasible in practice given the challenges in the operation of the AUV (limited endurance, complex deployment and retrieval procedures, etc...).

Author Response

Hello, thank you for your review and your comments.
I took into account your two suggestions and I added several elements :

- Concerning the sensors used by the AUV during the simulations, they were originally described in the section "4.1.2. The autonomous underwater vehicle", but I added additional elements in the section "6.2. Our main contributions" in order to make it clearer. The new elements are in the paragraph "Reducing the size of the state vector" and go from line 1100 to line 1110 (I am referring to the line numbers found on the right, in the PDF generated by latex). New elements about sensor fusion were also added in the section 4.1.2, from line 486 to line 489.
Moreover, we focused our work on the guidance and control algorithms, assuming that the navigation problem is resolved, as mentionned in the lines 223 and 224, and from line 548 to line 553.

- Concerning the applicability of our controller in a real setting, some elements about that were originally described in the section "7. Conclusions and openings". However, we added additional elements and new ideas from line 1154 to 1177.

Round 2

Reviewer 1 Report

The authors have accounted for my concern.